The biochemical and metabolic profiles of dairy cows with mycotoxins-contaminated diets

Wang Qian
Zhang Yangdong
Zheng Nan
Zhao Shengguo
Li Songli
Wang Jiaqi wangjiaqi@caas.cn jiaqiwang@vip.163.com
Chinese Academy of Agricultural Sciences, State Key Laboratory of Animal Nutrition, Institute of Animal Science , Beijing , People’s Republic of China
Chinese Academy of Agricultural Sciences, Key Laboratory of Quality & Safety Control for Milk and Dairy Products of Ministry of Agriculture and Rural Affairs, Institute of Animal Sciences , Beijing , People’s Republic of China
Chinese Academy of Agricultural Sciences, Laboratory of Quality and Safety Risk Assessment for Dairy Products of Ministry of Agriculture and Rural Affairs, Institute of Animal Sciences , Beijing , People’s Republic of China
Tiessen Axel
Electronic publication date: 2020 Mar 26
Publication date: 2020
Volume: 8
Electronic Location ID: e8742
Received 2019 Jan 17; Accepted 2020 Feb 13
Copyright: ©2020 Wang et al.
Copyright year: 2020
Copyright holder: Wang et al.
License: This is an open access article distributed under the terms of the Creative Commons Attribution License, which permits unrestricted use, distribution, reproduction and adaptation in any medium and for any purpose provided that it is properly attributed. For attribution, the original author(s), title, publication source (PeerJ) and either DOI or URL of the article must be cited.
License URL: https://creativecommons.org/licenses/by/4.0/

Keywords: Biochemical, Cottonseed, Dairy cow, Metabolomic, Mycotoxin

Funding: The National Key Research and Development Program 2017YFD0500502 Agro-scientific Research in the Public Interest 201403071 Modern Agro-Industry Technology Research System of the P.R. of China CARS-36 Introduce International Advanced Agriculture Science and Technology Plan 2016-X37 This work was supported financially by the National Key Research and Development Program (2017YFD0500502), the special Fund for Agro-scientific Research in the Public Interest (201403071), the Modern Agro-Industry Technology Research System of the P.R. of China (CARS-36), and the Introduce International Advanced Agriculture Science and Technology Plan (2016-X37). The funders had no role in study design, data collection and analysis, decision to publish, or preparation of the manuscript.

==============================
Background

Previous studies on the effects of mycotoxins have solely focused on their biochemical profiles or products in dairy ruminants. Changes in metabolism that occur after exposure to mycotoxins, as well as biochemical changes, have not been explored.

Methods

We measured the biochemical and metabolic changes in dairy cows after exposure to mycotoxins using biochemical analyses and nuclear magnetic resonance. Twenty-four dairy cows were randomly assigned to three different treatment groups. Control cows received diets with 2 kg uncontaminated cottonseed. Cows in the 50% replacement group received the same diet as the control group, but with 1 kg of uncontaminated cottonseed and 1 kg of cottonseed contaminated with mycotoxins. Cows in the 100% replacement group received the same diet as the control, but with 2 kg contaminated cottonseed.

Results

The results showed that serum γ-glutamyl transpeptidase and total antioxidant capacities were significantly affected by cottonseed contaminated with mycotoxins. There were also significant differences in isovalerate and NH3-N levels, and significant differences in the eight plasma metabolites among the three groups. These metabolites are mainly involved in amino acid metabolism pathways. Therefore, the results suggest that amino acid metabolism pathways may be affected by mycotoxins exposure.

Introduction

Mycotoxins are toxic secondary metabolites produced mainly by pathogenic molds that infect plants or crops, and by fungi in the genera Aspergillus, Fusarium, and Penicillium (Fink-Gremmels, 2008). Several hundred mycotoxins from animal feed have been identified and different studies have shown that most feed samples were infected with at least one mycotoxin (Rodrigues & Naehrer, 2012). The main types of mycotoxins that are extremely harmful to animals and their products are aflatoxins, deoxynivalenol (DON), zearalenone (ZEA), T-2 toxin, fumonisin, and ochratoxin A (Fink-Gremmels, 2008; Rodrigues & Naehrer, 2012).

Because of the potential hazards of mycotoxins, the effect of mycotoxins on dairy animal health is a major concern (Fink-Gremmels, 2008). Dairy cows may be regularly exposed to a mixture of mycotoxins derived from different ingredients in their diets. However, previous studies focused on investigating basic blood biochemical parameters and animal performance suggest that dairy cows are more unresponsive to mycotoxins than monogastric livestock because mycotoxins are readily degraded by rumen microbes (Dänicke et al., 2010; Pier, 1992; Santos & Fink-Gremmels, 2014). This was supported by the result that milk production tended not to decrease in lactating ewes fed diets contaminated with aflatoxin B1 (AFB1) (Battacone et al., 2009). Additional data has found that even calves are able to tolerate moderate feeding with different amounts of grains that are naturally contaminated with Fusarium toxins (Martin et al., 2010). Nevertheless, it is possible that dairy ruminants are more vulnerable to mycotoxins under high pressure of milk production (Jouany, Diaz & Diaz, 2005; Santos & Fink-Gremmels, 2014). Previous studies have demonstrated that exposure to AFB1 or mixed mycotoxins can change the antioxidant states of dairy cows and goats (Huang et al., 2018; Xiong et al., 2015), and it is possible that these changes may be reflected at minor physical levels. Thus, the effects of mycotoxins on animal performance still require further investigation.

Some studies have examined the effects of mycotoxins from the perspectives of molecular biology and metabolomics. In particular, it has been shown that exposure to AFB1 can influence the Nrf2 signaling pathway via mitochondrial reactive oxygen species (ROS)-dependent signaling pathways, specifically by impairing mitochondria functions and activating the generation of ROS (Liu & Wang, 2016). Mycotoxins can thus impair metabolic status, so it is necessary to investigate the dangers of mycotoxin contamination. Nuclear magnetic resonance (NMR) spectroscopy is a relatively simple method that can provide rich information about biochemical processes that are sensitive to the dynamic metabolic status of the organism (Maher & Rochfort, 2014; Nicholson & Wilson, 2003). Many metabolomics studies in different organisms have been conducted using NMR after exposure to mycotoxins (Cheng et al., 2017; Liu et al., 2013; Zhang et al., 2011). AFB1 exposure can significantly increase glucose and fatty acid levels, but according to NMR analyses, it can reduce the concentrations of lactate, ketone bodies, and amino acids in the serum (Cheng et al., 2017). Specific blood metabolites of dairy goats were induced by AFB1 exposure, indicating that endogenous metabolic alterations occur after exposure to mycotoxins. Therefore, changes in blood metabolic substrates of dairy ruminants can reflect physical conditions when animals are affected by internal or external factors (Cheng et al., 2017; Fink-Gremmels, 2008; Huang et al., 2018).

Despite several related studies, little research has been conducted on the potential metabolic signatures of dairy cows when exposed to mycotoxins. In the present study, we assessed biochemical parameters that reflect basic physiological conditions and changes in biomarkers based on plasma metabolomics using NMR investigation.

Materials and Methods

Preparation of diets contaminated with mycotoxins

Mycotoxins were cultivated and enriched on cottonseed contaminated with toxigenic fungi. We added water at a ratio of 30% relative to the weight of the uncontaminated cottonseed, and uniformly mixed the water and cottonseed. The wet cottonseed was then covered and placed in a relatively closed environment in order to facilitate the growth of fungi and accumulation of mycotoxins in the mixture. We measured the concentration of mycotoxins every 3 days (the cultivation period was set at 15 days, based on pilot experiments). When the AFB1 concentration in the cottonseed reached a suitable level, we dried the cottonseed out of the sunlight to stop the accumulation of mycotoxins.

Determination of mycotoxins in contaminated cottonseed

An AFB1 concentration of 20 µg/kg in the diet (dry matter) was the target concentration based on the limit specified by the European Union (EU) (European Commission, 2002). Additionally, the concentrations of two other mycotoxins, ZEA and DON, were also tested.

The concentrations of AFB1, ZEA, and DON were determined using the ROSA FAST Quantitative Test (Charm Sciences Inc, Lawrence, MA, USA). Five grams of contaminated ground cottonseed was mixed with 5 mL of methanol or water, and then shaken for 3 min. Next, 100 µL of the supernatant was removed and mixed with 1 mL of dilution buffer. Finally, 100 µL of the mixture was removed with a micropipette and spotted onto the test strip placed in an incubator for 5 min. The results were visually interpreted by two trained technicians using the ROSA Reader (ROSA Pearl Reader, Charm Sciences Inc., Lawrence, MA, USA). The results showed that 1 kg of contaminated cottonseed contained 20.08 µg AFB1 and 80.13 µg ZEA, but DON was not detected.

Animals, diets, and experimental design

The experiment was conducted at Ningxia Helan Sinofarm Dairy Farm (Ningxia, China) from December 17, 2015 to December 31, 2015 with an experimental period of 14 days. In total, 24 multiparous Holstein cows in late lactation (lactation length = 283 ± 22 days, milk yield = 21.1 ± 2.6 kg/day, parity = 2.5–3.5 times) were used in this study. The cows were raised in an open-sided, free stall barn with a sand bed, and were equipped with neck clips to allow a measured feed intake. Each of the three treatments had eight individuals randomly assigned to them. Cows in the control group received the uncontaminated diet (which included 2 kg of uncontaminated cottonseed). Cows in the 50% replacement group received the same diet as the control group, but the cottonseed component had 1 kg of uncontaminated cottonseed and 1 kg of cottonseed contaminated with mycotoxins. Cows in the 100% replacement group received the same diet as the control group, but the cottonseed component had 2 kg of cottonseed contaminated with mycotoxins (Table 1). Feedings occurred at 8:30, 16:30, and 00:30.

Table 1 Ingredients and mycotoxins composition of diets.1

Item	% DM	
Ingredient		
Oats hay	2.24	
Corn silage	45.57	
Alfalfa hay	8.07	
Soybean meal	4.86	
DDGS2	5.58	
Corn meal	11.88	
Wheat bran	2.24	
Extruded soy flour	0.56	
Corn bran shotcrete	4.59	
Cottonseed	4.48	
Mineral meal	0.22	
Diamond XP	0.13	
NaHCO3	0.54	
NaCl	0.07	
Premix	1.12	
Water	7.84	
Nutrients, % of DM		
Crude Protein	17.65	
Fat	4.15	
Non fiber carbohydrate	39.5	
Neutral detergent fiber	29.49	
Ca	0.8	
P	0.48	
Ash	9.27	
Energy (mcal/Kg)		
Metabolic energy	2.88	
Net Energy	1.67	
Mycotoxins (50% replacement), ppb3	
AFB1	20.08	
ZEA	85.13	
DON	ND4	
Notes.

1 Control group (0% of uncontaminated cottonseed replaced in diet); 50% replacement group (50% of the uncontaminated cottonseed replaced with cottonseed contaminated with mycotoxins).

2 Dried distillers grains with solubles.

3 ppb, µg/kg; AFB1, aflatoxin B1; ZEA, zearalenone; DON, deoxynivalenol.

4 ND, not detected.

All of the experimental procedures were approved by the Animal Care and Use Committee of the Institute of Animal Science, Chinese Academy of Agricultural Sciences (Protocol: IAS15020). The diets were formulated to meet and exceed the nutrient requirements for lactating Holstein cows (NRC, 2001). The ingredients, nutrients, and energy contents of the basal diets are listed in Table 1.

Sample collection and test methods

Diet samples

Diet samples were weighed, dried at 65 °C for 72 h, and stored at −20 °C for subsequent analyses. The nutrient, energy, and mineral ion indices were analyzed, including crude protein, fat, Ca, P, ash, non-fiber carbohydrate, and neutral detergent fiber (Table 1).

Blood parameters

Blood samples were collected from the jugular vein using vacutainer tubes (with and without anticoagulant) after milking on day 7. The samples were allowed to settle for about 1 h before centrifugation at 3,000 × g and 4 °C for 20 min to obtain serum, which was then frozen at −70 °C for subsequent analyses. The serum samples were analyzed to determine routine biochemical parameters using an Auto-Analyzer 7020 (Hitachi High-Technologies Corp., Tokyo, Japan) with colorimetric commercial kits (DiaSys Diagnostics Systems GmbH, Holzheim, Germany). Tested parameters included: alanine aminotransferase (ALT), aspartate aminotransferase (AST), γ-glutamyl transpeptidase (GGT), alkaline phosphatase (ALP), total protein, albumin, globulin, albumin/globulin ratio, urea, creatinine, uric acid, total bilirubin, direct bilirubin, indirect bilirubin, triglycerides, and total cholesterol. Another set of serum samples for determining immune and antioxidant indices was sent to Beijing CIC Clinical Laboratory (Beijing, China), where the concentrations of immunoglobulin M (IgM), IgA, and IgG were determined using bovine immunoglobulin ELISA kits (Shanghai Meilian BioTech Company, Shanghai, China). The methods used to determine the total antioxidant capacity (T-AOC), superoxide dismutase (SOD), glutathione peroxidase (GSH-Px), and malondialdehyde (MDA) levels were described in previous studies (Cheng et al., 2017; Xiong et al., 2015).

Concentrations of volatile fatty acids (VFAs) and NH3-N in rumen fluid

Rumen fluid was collected using an oral stomach tube about 1 hr after the morning feeding on day 7, as described in a previous study (Shen et al., 2012). The first 50–100 ml of flowing liquid was discarded to avoid contamination with saliva, and the oral stomach tube was washed twice using fresh water before the next sampling. The concentrations of rumen VFAs were determined using a gas chromatography system equipped with flame ionization (GC 6890N, Agilent, Wilmington, DE). The rumen liquid was deproteinized with 2 mL of 25% metaphosphoric acid and frozen before each VFA analysis. Helium was used as the carrier gas, and a mixture of six species of VFAs (Supelco Inc., Bellefonte, PA, USA) was used as the identification standard. The initial and final temperatures in the oven were 55 °C and 195 °C, respectively, and the detector and injector were set to 250 °C. The samples of deproteinized ruminal fluid were neutralized with potassium hydroxide solution and then centrifuged at 1,200 × g for 10 min. Ruminal NH3-N was analyzed with a UV-2000 ultraviolet visible spectrophotometer (Unico Instrument Co. Ltd, Shanghai, China).

Metabolomic analysis of plasma by 1H NMR

Deuterium oxide (D2O) and deuterated chloroform were purchased from Cambridge Isotope Laboratories, Inc. (Tewksbury, MA, USA). In addition, 3-(trimethylsilyl) propionic-2,2,3,3,d4 propionic acid sodium salt was purchased from Merck Inc. (Kirkland, QC, Canada). HPLC-grade methanol, methyl tert-butyl ether, water, formic acid, and ammonium formate were purchased from Merck (Darmstadt, Germany).

Twenty-four frozen plasma samples were thawed at room temperature and 200 µl of each sample was mixed with 400 µl of buffer (45 mM NaH2PO4/K2HPO4; 0.9% NaCl; pH: 7.4; 50% D2O). After shaking and mixing, the sample was centrifuged at 4 °C and 16, 099 ×  g for 10 min. The supernatant was placed in a 5-mm NMR tube for examination. A Bruker AVIII 600 MHz NMR (proton resonance frequency =600.13 MHz, ultra-low temperature probe; Bruker BioSpin GmbH, Rheinstetten, Germany) system was used for 1H NMR analysis of the blood samples.

In order to attenuate the NMR signals to avoid the influence of macro-molecules, a water-pre-saturated Carr-Purcell-Meiboom-Gill (CPMG) pulse sequence was employed (recycle delay- 90°-(τ-180°-τ)n- acquisition). A one-dimensional CPMG pulse sequence with pre-saturated pressurized water was used to detect the small molecule metabolites in each sample. The experimental parameters were as follows: spectral width, 12,000 Hz; waiting time, 2 s; mixing time, 100 ms; and sampling number, 32K. NMR spectra were manually phased, baseline corrected, and referenced to TSP (CH3, δ0.00) using Bruker Topspin 3.0 software (Bruker GmbH, Karlsruhe, Germany). NMR spectra were visually inspected using Amix 3.9.13 (Bruker, Biospin, Italy). Finally, the NMR spectra were integrated over the 9.0–0.5 ppm range using an interval of 0.002 ppm and the water peak (δ 5.20–4.20) was removed.

Data analysis

Blood biochemical parameters, antioxidant and immune indices, rumen fluid VFA levels, and NH3-N data were analyzed using analysis of variance in SPSS Statistics, and post-hoc tests were conducted (IBM SPSS Statistics v19.0, SPSS Inc., Chicago, IL, USA). The statistical models included treatments as the fixed effects and cows within treatment as the random effect. The data for food intake and milk yield before the first day of the treatment period were used as a covariate in the statistical analysis. Tukey’s multiple comparisons adjustment was used to determine significant differences between least squares means. Statistically significant differences were accepted at P < 0.05.

The 1H NMR spectra were subjected to Fourier transformation, phase adjustment, baseline correction, and calibration using MestReNova V7.0 software (Mestrelab Research SL, Santiago de Compostela, Spain). To improve the signal-to-noise ratio, all of the spectra were multiplied by an exponential function of 1 Hz before Fourier transformation. The 1H NMR spectra were referenced to an internal lactic acid CH3 resonance at 1.33 µg/kg. The 1H NMR spectra were segmented into consecutive non-overlapping regions comprising 0.002 µg/kg chemical shift “bins” between 0.5 and 9.0 µg/kg. The residual water peak at 4.18–6.70 µg/kg was removed from the data. The normalized data were analyzed by multivariate analysis using SIMCA-P+ software (V11.0 Umetrics AB, Umea, Sweden). First, the 1H NMR spectra were analyzed using principal component analysis (PCA) based on mean center scaling to reflect the overall differences. Next, the spectra were analyzed by supervised methods of partial least-squares discriminant analysis (PLS-DA) and orthogonal partial least-squares discriminant analysis (OPLS-DA) (Lundstedt, Trygg & Holmes, 2007). The quality of each model was determined based on the goodness of fit parameter (R2) and a goodness of prediction parameter (Q2) (Eriksson et al., 2006). The statistical significances of differences in the metabolite concentrations and appropriate correlation coefficients were determined by OPLS-DA.

Results

Blood biochemical, antioxidant, and immune parameters

There were no significant differences in most of the serum parameters (Table 2). However, there was a significant difference in GGT between the control group and the 50% replacement group (P < 0.05), although there was no significant difference in GGT between the control group and the 100% replacement group. In addition, the difference in T-AOC between the control group and the 100% replacement group reached a very significant level, but there was no significant difference between T-AOC in the control group and the 50% replacement group.

Table 2 Effects of cottonseed contaminated with mycotoxins1 on the serum biochemical, antioxidant, and immune indices of dairy cows.

Item2	Control	50%replacement	100% replacement	SEM	P-value3	
ALT (U/L)	29.75	29.75	29.86	1.05	0.99	
AST (U/L)	72.75	71.00	77.50	3.03	0.68	
AST/ ALT	2.52	2.41	2.62	0.11	0.78	
GGT (U/L)	43.09a	29.30b	34.45a,b	2.16	0.02	
ALP (U/L)	90.31	67.61	71.19	12.36	0.74	
TP (g/L)	73.96	73.53	73.53	0.97	0.98	
ALB(g/L)	36.44	35.49	36.14	0.45	0.69	
GLOB(g/L)	37.53	38.04	37.39	1.08	0.97	
A/G	0.98	0.94	1.01	0.03	0.72	
UREA(mmol/ml)	3.10	3.45	3.30	0.11	0.44	
CR (µmol/L)	71.16	67.50	70.36	2.13	0.78	
UA(µmol/L)	25.85	28.56	30.49	1.94	0.64	
TBil(µmol/L)	11.05	9.82	9.82	0.53	0.46	
DBil(µmol/L)	2.28	2.03	2.40	0.09	0.21	
IBiL(µmol/L)	8.78	7.79	9.00	0.45	0.53	
TG(mmol/ml)	0.05	0.05	0.06	0.004	0.74	
TC(mmol/ml)	6.03	5.95	6.71	0.26	0.46	
GSH-PX (U/ml)	669.0	661.5	687.8	12.48	0.70	
MDA(nmol/ml)	12.42	12.77	8.02	1.18	0.19	
SOD(U/ml)	106.9	108.9	107.4	0.81	0.57	
SOD/ MDA	10.32	11.8	15.63	1.31	0.24	
T-AOC(U/ml)	2.40a	2.39b	3.52b	0.26	0.009	
IgG (µg/ml)	13.67	18.75	13.75	2.13	0.56	
IgA (ng/ml)	59.43	50.82	53.57	4.46	0.43	
IgM (ng/ml)	22.40	23.42	22.30	1.95	0.93	
Notes.

1 Control group (uncontaminated cottonseed), 50% replacement group (50% of the uncontaminated cottonseed replaced with cottonseed contaminated with mycotoxins), and 100% replacement group (100% of the uncontaminated cottonseed replaced with cottonseed contaminated with mycotoxins).

2 ALT, alanine aminotransferase; AST, aspartate aminotransferase; GGT, -glutamyl transpeptidase; ALP, alkaline phosphatase; TP, total protein; ALB, albumin; GLOB, globulin; A/G, albumin/ globulin; UREA, urea; CR, creatinine; UA, uric acid; TBil, total bilirubin; DBil, directed bilirubin; IBiL, indirect bilirubin; TG, total triglyceride; TC, total cholesterol; GSH-PX, glutathione peroxidase; MDA, malondialdehyde; T-AOC, total antioxidant capacity; SOD, superoxide Dismutase; IgG, immunoglobulin G; IgA, immunoglobulin A; IgM, immunoglobulin M.

3 Probability associated with the F-test based on differences between treatments.

ab Means in the same row with different superscripts are significantly different (P < 0.05) according to Tukeys test.

Rumen function

The ruminal concentrations of VFAs and NH3-N were used as indicators of rumen fermentation and the effects of dietary treatments. We found that mycotoxins significantly increased (P < 0.05) the rumen NH3-N concentration in the 100% replacement group but not in the 50% replacement group (Fig. 1A). In addition, the contaminated cottonseed did not affect the total amount of rumen VFAs (Fig. 1B). Interestingly, the different levels of added contaminated cottonseed did not affect the concentrations of acetate, propionate, butyrate, and valerate, but the contaminated cottonseed had significant effects on isovalerate (P < 0.05) (Table 3). The isovalerate concentrations in the control group differed from those in the 50% replacement group (P < 0.05), but there were no significant differences between the control group and the 100% replacement group.

Metabolomic profiling of plasma based on NMR analysis

Representative 1H NMR spectra (δ 9.0–0.5) of the plasma samples obtained from the control group, 50% replacement group, and 100% replacement group are shown in Fig. 2. These metabolites were identified and compared based on data obtained from a previous study (Nicholson & Wilson, 2003) and the ChenomX spectral database (Edmonton, AB, Canada). The data were subsequently analyzed using multivariate statistics, i.e., PCA, PLS-DA, and OPLS-DA.

Figure 1 Effects of cottonseed contaminated with mycotoxins on the concentration of NH3-N (A) and total VFA (Volatile Fatty Acid) (B) in the cows rumen.

Notes: Control group = uncontaminated cottonseed; 50% replacement group = 50% of the uncontaminated cottonseed was replaced with cottonseed contaminated with mycotoxins; and 100% replacement group = 100% of the uncontaminated cottonseed was replaced with cottonseed contaminated with mycotoxins.

Table 3 Effects of cottonseed contaminated with mycotoxins1 on volatile fatty acid concentrations in the cow rumen.

Item (mmol/L)	Control	50%replacement	100%replacement	SEM	P-value2	
Acetate	65.55	62.72	60.78	1.08	0.19	
Propoinate	22.49	22.76	22.03	0.41	0.78	
Acetate/Propoinate	2.93	2.77	2.77	0.06	0.43	
Isobutyrate	0.66	0.72	0.75	0.02	0.27	
Butyrate	12.73	12.31	12.21	0.36	0.84	
Isovalerate	1.27a	1.61b	1.29ac	0.05	0.008	
Valerate	1.39	1.51	1.31	0.06	0.21	
Notes.

1 Control group (uncontaminated cottonseed), 50% replacement group (50% of the uncontaminated cottonseed replaced with cottonseed contaminated with mycotoxins), and 100% replacement group (100% of the uncontaminated cottonseed replaced with cottonseed contaminated with mycotoxins).

2 Probability associated with the F-test based on differences between treatments.

abc Means in the same row with different superscripts are significantly different (P < 0.05) according to Tukeys test.

Figure 2 Representative 600 MHz 1D NOESY 1H-NMR spectra (δ0.5–5.5 and δ5.5–9.0) from cows plasma.

(A) Control group, (B) 50% replacement group (50% of the uncontaminated cottonseed was replaced with cottonseed contaminated with mycotoxins), and (C) 100% replacement group (100% of the uncontaminated cottonseed was replaced with cottonseed contaminated with mycotoxins). The δ5.5–9.0 region was magnified 16 times relative to the corresponding δ0.5–5.5 region for clarity. Abbreviations: Glu: Glutamate; NAG, N-acetyl glycoprotein signals; L1, LDL (Low Density Lipoprotein), CH3–(CH2)n-; L2, VLDL (Very Low Density Lipoprotein), CH3–(CH2)n-; L3, LDL, CH3–(CH2)n-; L4, VLDL, CH3–(CH2)n-; L5, VLDL, −CH2; L6, lipid, −CH2–CH =CH-; L7, lipid, −CH2–C =O; L8, lipid, =CH-CH2–CH=.

To obtain an overview of the data, PCA was performed to identify the two principal components (PC1 and PC2), where the cumulative variance contribution rate of the 1H NMR spectra was 90.9% (PC1 =81.5% and PC2 =9.4%) (Fig. 3A). However, there was some overlap and one or two discrete values. Figs. 3B–3D show that the control and the 50% replacement group, the control and the 100% replacement group, and the 50% replacement and 100% replacement group, respectively, were not well separated from each other, so it was necessary to use supervised methods for further analysis. OPLS-DA increased the number of principal components until the variance explained by the model (R2) or the predictive variance of the model (Q2) reached 2%. The validity of the model was established by conducting cross and permutation tests (200 times). After this procedure, OPLS-DA score plots (A, D, and G) derived from 1H NMR spectra for plasma and the corresponding coefficient loading plots (B, C, E, F, H, and I, in which, B, E and H are images magnified 20 times) were shown in Fig. 4, the control and 50% replacement groups were discriminated with R2 X = 32.5% and Q2 =0.4 (Fig. 4A), the control and 100% replacement groups with R2 X = 29.3% and Q2 =0.374 (Fig. 4D), and the 50% and 100% replacement groups with R2 X = 26.5% and Q2 =0.316 (Fig. 4G).

Figure 3 PCA plot based on the 1H NMR spectra for plasma obtained from different groups.

(A) Score plot for the PCA model of the control, 50% replacement group, and 100% replacement group; (B) score plot for the PCA model of the control and 50% replacement group; (C) score plot for the PCA model of the control and 100% replacement group; (D) score plot for the PCA model of the 50% replacement group and 100% replacement group. Control (I), black square; 50% replacement group (50% of the uncontaminated cottonseed was replaced with cottonseed contaminated with mycotoxins) (II), red spot; 100% replacement group (100% of the uncontaminated cottonseed was replaced with cottonseed contaminated with mycotoxins) (III), blue diamond.

Figure 4 OPLS-DA score plots (A, D, and G) derived from 1H NMR spectra for plasma and the corresponding coefficient loading plots (B, C, E, F, H, and I, in which, B, E and H are images magnified 20 times).

Control (I), black square; 50 % replacement group (50% of the uncontaminated cottonseed was replaced with cottonseed contaminated with mycotoxins), (II), red spot; 100% replacement group (100% of the uncontaminated cottonseed was replaced with cottonseed contaminated with mycotoxins), (III), blue diamond. The color map shows the significant variations in metabolites between the two classes. Peaks in the positive direction indicate that metabolites were more abundant in the groups in the positive direction of the first principal component. Thus, metabolites that were more abundant in the three groups in the negative direction of the first primary component were shown as peaks in the negative direction.

The plasma spectra mainly contained signals from glycoproteins, glucose, amino acids, creatinine, and citrate metabolites. The plasma concentrations of the eight metabolites differed significantly among the three groups according to the results obtained by 1H NMR analysis (Table 4), particularly amino acids comprising alanine, lysine, glutamine, and glycine, the carboxylic acids creatinine and citrate, glucose, and O-acetyl-glycoprotein. Using the metabolomics view map derived from pathway topology analysis (MetaboAnalyst 3.0), we found the enrichment analysis and path impact values of different pathways. The pathways ranked in the top three were aminoacyl-tRNA biosynthesis, nitrogen metabolism and alanine, and aspartate and glutamate metabolism (Fig. 5).

Discussion

Biochemical parameters in blood

Many components of blood can reflect physiological functions as well as immune and antioxidant activities in the body. For example, GGT, AST, ALT, and ALP are used as liver function indicators (Edrington, Harvey & Kubena, 1994; Xiong et al., 2015). We found no significant differences in most of the serum parameters in the control and two treatments. However, two variables were affected by the intake of mycotoxins. The blood concentration of GGT is considered an indicator of liver function in ruminants (Osorio et al., 2014; Xiong et al., 2015). Thus, in the present study, the significant difference in GGT between the control group and 50% replacement group suggests that liver function was affected by contaminated cottonseed, although it is unclear why the levels did not change significantly in the 100% replacement group. Mycotoxins can influence liver function, but its effects fluctuate according to toxin dose and duration. A daily intake of 3-128 µg/day of pure AFB1 for one week did not alter the activities of several enzymes related to liver function in dairy sheep (Battacone et al., 2005), but the blood activity level of ALT increased significantly in dairy ewes after the intake of a high dose of AFB1 (128 µg/day) for two weeks (Battacone, 2003). In lambs fed 2.5 mg/kg AFB1 for 35–67 days, the serum GGT and AST levels increased significantly (Edrington, Harvey & Kubena, 1994). However, in a different study, AFB1 did not affect the plasma concentrations of AST, ALT, GGT, nor ALP in dairy cows (Xiong et al., 2015).

Table 4 OPLS-DA coefficients derived from the NMR data of plasma metabolites obtained from (A) control, (B) 50% replacement group (50% of the uncontaminated cottonseed replaced with cottonseed contaminated with mycotoxins) and (C) 100% replacement group (100% of the uncontaminated cottonseed replaced with cottonseed contaminated with mycotoxins).

Metabolitesb	ra	
	A vs. B (df = 7)	A vs. C (df = 7)	B vs. C (df = 7)	
Ala(Alanine): 1.48(d)	–	−0.669	0.763	
Lys(Lysine): 1.49(m), 1.74(m), 1.91(m)	0.743	–	0.697	
Gln(Glutamine): 2.12(m), 2.42(m)	–	–	0.733	
OAG(O-acetyl-glycoprotein): 2.13(s)	–	–	0.668	
Cit(Citrate): 2.54(d), 2.68(d)	–	−0.748	0.695	
Cr(Creatine): 3.03(s),3.93(s)	–	–	0.674	
Gly(Glycine): 3.56(s)	–	–	0.731	
Glucose: 3.42(t), 3.54(dd), 3.71(t), 3.73(m), 3.84(m), 3.25(dd), 3.41(t), 3.46(m), 3.49(t), 3.90(dd), 4.65(d), 5.23(d)	–	–	0.768	
Notes.

a Correlation coefficients where positive and negative signs indicate positive and negative correlations between the concentrations, respectively. Correlation coefficients of |r| > 0.666, 0.707, or 0.755 were used as the cutoff values for significant differences based on the discrimination significance at the P = 0.05 level, where df (degrees of freedom) = 7, 6, or 5; “-” indicates that the correlation coefficient |r| was less than 0.666, 0.707, or 0.755.

b Multiplicity: s, singlet; d, doublet; t, triplet; q, quartet; dd, doublet of doublets; m, multiplet; br, broad resonance.

Figure 5 Metabolome view map showing the matched pathways according to the Pvalues obtained from pathway enrichment analysis and the pathway impact values produced by pathway topology analysis.

Aminoacyl-tRNA biosynthesis (a); nitrogen metabolism (b); alanine, aspartate, and glutamate metabolism (c). The x-axis represents the pathway impact and the y-axis represents pathway enrichment. In the map, dots with larger sizes and darker colors represent higher pathway enrichment and higher pathway impact values, respectively.

A previous study showed a slightly stronger reaction to 20 µg/kg AFB1 than 40 µg/kg AFB1, which was reflected by the MDA levels (Xiong et al., 2015). Aspartate aminotransferase (AST), one of the most important transaminases, is an indicator of liver function in clinical medicine used to judge whether the liver is damaged. Data has shown that the group fed a meal without mycotoxins, the group fed AFB1, and the group fed AFB1 mixed with ZEA had similar serum AST levels, but the group fed AFB1 mixed with OTA showed significantly higher levels than that of the three groups. Moreover, the group fed AFB1 mixed with OTA and ZEA had significantly higher levels than that of the abovementioned four groups (Huang et al., 2018). These results may hint that mycotoxin type and amount are key factors that can affect animal health performances.

The serum concentrations of IgM, IgA, and IgG in the present study did not change significantly, suggesting that the contaminated cottonseed did not affect the immune functions of the dairy cows. Similar results were found in other studies (Korosteleva, Smith & Boermans, 2007; Xiong et al., 2015), but it is possible that the response to mycotoxins may be reflected in other immune factors. Additional parameters were not tested in the current study, and immune function measures are required for further comprehensive analysis.

Previous studies have shown that mycotoxins can influence animal oxidant stress levels. SOD and MDA are used as indices of the degree of lipid peroxidation, and the SOD/MDA ratio can reflect free radical-induced lipid peroxidation and the scavenging rate (Surai, 2002; Xiong et al., 2015). For example, mycotoxins increase the MDA concentration, but decrease the SOD concentration in dairy goats (Huang et al., 2018). In the present study, we found that the 100% replacement group had a lower concentration of serum T-AOC compared with the other two groups. Since T-AOC is an indicator of overall biochemical antioxidant capacity, it reflects the systemic ability of antioxidative enzymes and non-antioxidative enzymes to compensate for external stress and the capacity to clear free radicals (Surai, 2002). Therefore, lower T-AOC levels may indicate that contaminated cottonseed affected the overall antioxidant capacity of dairy cows. A previous study showed that the reaction to 20 µg/kg AFB1 was slightly stronger than the reaction to 40 µg/kg AFB1, which was reflected by the MDA level and SOD/MDA ratio (Xiong et al., 2015). It should be noted that subtle changes may affect overall antioxidant capacity in response to toxins (but not at a significant level), which may be sporadically detected in terms of other oxidative indicators, such as SOD, MDA, and GSH-Px.

Rumen function

Feedstuffs (e.g., plant constituents, fiber, and cellulose) are fermented in the rumen of dairy cows with the aid of rumen microbes, bacteria, protozoa, and fungi. Evidence suggests that rumen microbiota can protect animals by binding, deactivating, and degrading toxic molecules (Fink-Gremmels, 2008).

Rumen microorganisms convert carbohydrates into VFAs, and these organic acids are absorbed through the gastrointestinal tract into the circulatory system to reach different tissues. VFAs are consequently an essential energy resource for ruminants. The rumen concentration of VFAs has been used as an indicator of rumen fermentation and the effects of dietary treatments (Hall et al., 2015; Xiong et al., 2015). In the present study, the different levels of contaminated cottonseed affected isovalerate but did not affect the concentrations of total VFAs, acetate, propionate, butyrate, isobutyrate, and valerate (Fig. 1 and Table 3). A recent study suggested that the concentration of total VFAs in rumen is not an appropriate indicator of ruminal fermentation or microbial product formation because of the high daily variation in the amount of rumen fluid (Hall et al., 2015), and this finding was supported by another study (Xiong et al., 2015). Thus, it may be more useful to investigate the levels of specific types of VFAs. Isovalerate, isobutyric acid, and 2-methyl butyrate belong to the family of isoacids. Previous studies have indicated that isoacids promote the growth of anaerobic bacteria in the rumen fluid and degrade fiber in the rumen (Liu et al., 2014). In vitro analyses of rumen fermentation have shown that isoacids can promote the growth of micro-organisms and the degradation of fiber after adding a mixture of isobutyrate, 2-methyl butyrate, and isovalerate (Gunter et al., 1990). Thus, the significant differences in isovalerate levels in the present study may indicate that rumen fermentation is influenced by contaminated cottonseed. However, similar experiments were inconclusive in many previous animal studies. A similar study found that mixed mycotoxins significantly increased NH3-N and total VFA concentrations (Kiyothong et al., 2012). Another study found that the acetate:propionate ratio in the rumen of dairy cows was influenced by supplementation with 20 µg/kg AFB1 (Xiong et al., 2015). Other studies found no evidence that feeding with isoacids improved digestibility in the rumen (Gunter et al., 1990), with one study in particular showing that supplementation with isoacids does not change digestibility of dry material, fiber, and crude protein in steers (Mccollum, Kim & Owens, 1987).

In the present study, the significant difference in rumen isovalerate between the control and 50% replacement groups may suggest that rumen fermentation was affected by contaminated cottonseed. It is unclear why the levels did not change significantly in the 100% replacement group. The experimental data of isovalerate addition showed that the low, middle and high doses (100, 200, and 300 mg per DMI) were not consistent with our expectations. The rumen isobutyrate content in the middle dose was the highest. Rumen isobutyrate levels in the low dose group were the same as in the control group (Liu et al., 2009). Further research is needed to clarify the relationships between mycotoxins, diet, and the concentrations of specific VFA in the rumen.

A suitable concentration of NH3-N is an important mediator in rumen nitrogen metabolism. We found that mycotoxins significantly increased the NH3-N concentration in the rumen in the 100% replacement group (Fig. 1). The most suitable ruminal NH3-N concentration for growth ranges between 5 and 28 mg/100 mL (Wanapat & Pimpa, 1999). However, NH3-N concentration in the 100% replacement group went beyond the upper limit of this range, suggesting that the microbial protein decomposition capacity could be imbalanced (Wanapat & Pimpa, 1999). The contaminated cottonseed may have affected the microbial degradation of proteins and the synthesis of ammonia, for two possible reasons. First, isovalerate could have promoted the utilization of microbial nitrogen to produce NH3-N, thereby leading to increased microbial protein synthesis (Allison, 1969). Second, the rumen NH3-N concentration was reduced due to the decreased rate of protein degradation. Our results showed that the decrease in isovalerate may have led to the reduced utilization of NH3-N by rumen microbes, or some other mechanism related to in vivo nitrogen retention may have been partially blocked in the rumen (Felix, Cook & Huber, 1980).

Analysis of plasma metabolites and metabolic pathways

Feedstuffs can have important effects on the performances of dairy cows, and metabolomics can help us accurately understand the causes of metabolic changes and their subsequent effects (Cheng et al., 2017; Sundekilde et al., 2013). In the present study, eight plasma metabolites differed significantly among the three groups. These metabolites were mainly involved in the metabolic pathways related to aminoacyl-tRNA biosynthesis, nitrogen metabolism, and alanine, aspartate, and glutamate metabolism (Table 4 and Fig. 5). Interestingly, four amino acids included in the eight metabolites differed significantly in the present study, indicating that amino acid metabolism was affected by the mycotoxins. Previous studies have shown that mycotoxins can influence energy expenditure and protein metabolism (Sun et al., 2014; Sundekilde et al., 2013). Generally, the balance of amino acids is regulated by the relative rates of synthesis and degradation. Studies suggest that acylated amino acids to tRNA are the immediate precursors to protein synthesis and that each amino acid is catalyzed by a specific synthase that combines with the corresponding tRNA (Davis et al., 1999; Martin et al., 1977). However, the aminoacyl-tRNA concentration is very low and extremely unstable within cells and tissues, and previous studies have shown that tRNA plays an important regulatory role in the gene expression process within the dairy cow’s mammary gland (Bauman et al., 2006; Mackle et al., 2000; Wang et al., 2014). When an organism encounters external stress, the unloaded tRNA can act as an effector for the overall gene expression levels in the cell, thereby allowing the organism to cope with an adverse environment. tRNA can be used for this purpose in yeast and some mammalian cells. The nucleus monitoring system can continuously monitor the integrity of tRNA, and in the absence of nutrition, the retrograde transport of tRNA into the nucleus can effectively reduce the level of protein synthesis (Davis et al., 1999; Martin et al., 1977). Previous studies have suggested that investigations of the effects of mycotoxins on animal metabolism should focus specifically on amino acid metabolism.

Nitrogen metabolism (nitrogen balance) mediates the relationship between nitrogen intake and excretion. Previous studies have indicated that feed constituents can influence nitrogen metabolism in dairy cows (Cantalapiedra-Hijar et al., 2014; Otto et al., 2003). Amino acids are important nutrients in cow milk (Sundekilde et al., 2013) and a major aspect of ruminant protein nutrition research involves studying the digestion and absorption of feed proteins (Cantalapiedra-Hijar et al., 2014; Castillo et al., 2001). Nitrogen balance can directly affect health status and pasture waste management (Arriaga et al., 2009). In the present study, the nitrogen metabolism pathway was significantly affected, probably due to the effects of mycotoxins on the synthesis and degradation of amino acids. The involved mechanism may influence glucogenic nutrients and the utilization of amino acids. The amino acids that remain in the body after utilization by the liver and intestines are transferred to the peripheral organs and tissues or excreted as waste. Thus, the visceral organs may play a regulatory role in nitrogen metabolism in order to limit the systemic availability of absorbed amino acids to the peripheral tissues (Cantalapiedra-Hijar et al., 2014; Larsen et al., 2015).

Our results also illustrate that mycotoxins affect glucose metabolites in the blood, thereby influencing glucose metabolism. Previous studies have shown that mycotoxin-contaminated feed can affect glucose metabolism, specifically by decreasing the activities of intestinal glucose transporters (Bertrand & Applegate, 2013; Liu et al., 2013; Zhang et al., 2011). Glucogenic amino acids are involved in the regulation of glucose metabolism (Xu et al., 2008). Branched-chain amino acids may inhibit glycogenolysis in the liver and muscles, and can enhance the alanine-glucose and lactic acid-glucose cycles (Hayirli, 2006). Alanine, aspartate, and glutamate are glucogenic amino acids. Glutamate is a functional amino acid with important physiological regulatory functions (Duan et al., 2013; Zhang et al., 2013). We found that feed contaminated with mycotoxins had significantly affected alanine, aspartate, and glutamine metabolism pathways, indicating that body gluconeogenesis is also influenced by the mycotoxins.

In the present study, we found significant differences in the O-acetyl-glycoprotein, citrate, and creatinine concentrations among the three treatment groups. A previous study similarly showed that exposure to ZEA significantly elevated the plasma levels of glucose and O-acetyl glycoprotein (Liu et al., 2013). NMR has been used to identify the milk metabolites obtained from two dairy cow breeds (Danish and Jersey Holstein), and they found choline, creatinine, and citrate, which are also potential biomarkers (O’Sullivan et al., 2013; Pinotti, Baldi & Dell’Orto, 2002). Our findings suggest that O-acetyl-glycoprotein, citrate, and creatinine may be used as potential biomarkers of altered metabolism.

Conclusion

Diets containing cottonseed contaminated with mycotoxins significantly influenced the blood GGT content and T-AOC of dairy cows. Significant differences in isovalerate and NH3-N concentrations were also found between the control and treatment groups, indicating that the contaminated cottonseed may have influenced ruminal function. Moreover, among the three treatment groups, there were significant differences in the eight plasma metabolites mainly involved in the aminoacyl-tRNA biosynthesis, nitrogen metabolism, and alanine, aspartate, and glutamate metabolism pathways. These differences suggest that amino acid metabolism pathways may be important targets when investigating the effects of mycotoxin exposure in future research.

Supplemental Information

Data S1 Raw data

Click here for additional data file.

We sincerely thank Dr. Cheng, his graduate students, and the farm staff for their help in this experiment. We also sincerely thank the anonymous reviewers for their revision suggestions.

Additional Information and Declarations

Competing Interests

Author Contributions

Animal Ethics

Data Availability

The authors declare there are no competing interests.

Qian Wang, Yangdong Zhang and Jiaqi Wang conceived and designed the experiments, performed the experiments, analyzed the data, prepared figures and/or tables, authored or reviewed drafts of the paper, and approved the final draft.

Nan Zheng conceived and designed the experiments, authored or reviewed drafts of the paper, and approved the final draft.

Shengguo Zhao and Songli Li conceived and designed the experiments, prepared figures and/or tables, and approved the final draft.

The following information was supplied relating to ethical approvals (i.e., approving body and any reference numbers):

All experimental procedures were approved by the Animal Care and Use Committee of the Institute of Animal Science, Chinese Academy of Agricultural Sciences (Protocol: IAS15020).

The following information was supplied regarding data availability:

The raw data is available in a Supplementary File.

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
