# Peer review of "The biochemical and metabolic profiles of dairy cows with mycotoxins-contaminated diets"

_PeerJ, doi:10.7717/peerj.8742_

## Round 0.1 · original submission · Major Revisions

Dear author,

Your paper has been assessed by three reviewers and myself as academic Editor.

As you could see below, the manuscript needs a major revision.
Most importantly:

The experimental design needs to be improved. Please perform mycotoxin analyses in either blood, rumen, urine, or milk samples.
There are many error typing errors throughout the manuscript.
Revise by a native speaker to meet the professional standards of courtesy and expression for publication.
References should be in consistent format.
Misconceptions need to be corrected.
Statistical treatment of data needs to be more rigorous.
Please address all concerns of the reviewers and submit a revised version of the manuscript. Please include a detailed response to each reviewer.

I have been informed that PeerJ is now offering a language correction service for a fee. Please consider it before submitting the new version of the manuscript.

Reviewer 1 ·

Basic reporting

The manuscript from Qian Wang et al. reports the biochemical and metabolic profiles of dairy cows with mycotoxins-contaminated diets. Specifically, the cottonseed contaminated with mycotoxins was cultivated by promoting the growth of fungi, and used to feed multiparous Holstein cows with an experimental period of 14 days. The authors finally provided data of blood parameters, concentrations of volatile fatty acids (VFAs) and NH3-N in rumen fluid, metabolomic analysis of plasma by 1H NMR. Overall, the manuscript seems to be interesting that provides evidence supporting the amino acid metabolism pathways may be affected by exposure to mycotoxins. However, there are issues about the obtained data raising throughout the manuscript.

Experimental design

The authors gave the effort to describe the method of preparation of diets contaminated with mycotoxins and how to determine mycotoxins in contaminated cottonseeds. However, what is exactly fungal species using as “promoting the growth of fungi” (line 93) should be obviously stated. If this method will get an advantage compared to the use of purified mycotoxins should be addressed. Also, this research reported the biochemical and metabolic profiles of dairy cows under the exposure of combination of AFB1 and ZEA in contaminated cottonseeds, please discuss if there is any co-effect of both mycotoxins in the currently obtained data.

Validity of the findings

Line 233: “Table 5”. If it is Table 4 in this manuscript.
Line 316: “Table 5”. If it is Figure 5 in this manuscript.
Figure 5 is confusing. Please describe in detail about how to create Figure 5 with a specific definition of “the matched pathways according to the P-values obtained from pathway enrichment analysis and the pathway impact values produced by pathway topology analysis”.

Additional comments

There are many error typing throughout the manuscript. Also, the manuscript should be revised by a native speaker to meet the professional standards of courtesy and expression for publication.
References should be in consistent format.

Reviewer 2 ·

Basic reporting

The paper must be revised by an English editor.
The list of references is not very well prepared.

Experimental design

The experimental design seems to be flawed:
The authors failed to mention the composition of the diet that the cows were consuming previously, especially whether it included cottonseed. So, if the ruminal microbiota had not been exposed to any test ingredient, a minimum of about three weeks would be needed just to get the microorganisms adapted to the new feedstuff. Therefore fourteen days was obviously insufficient, moreover considering that both blood and rumen samples were obtained (only once) on day 7, why run a 14-day trial if samples were obtained only on day seven?
The measurement of the rumen VFA’s might be incomplete without assessing the total ruminal volume, which is usually quite variable between animals.
Although the authors state that the ruminal microbiota is capable of neutralizing the mycotoxins, It is worth noting that no mycotoxin analyses were performed in neither blood, nor rumen, urine, or milk samples. These analyses could yield important information on whether the compounds were in any way altered or destroyed by the microbiota.
I suppose the milk from the experimental cows fed the mycotoxins was destroyed, although this is not mentioned in the article.

Validity of the findings

The statement on line 203 is not what the data in table 2 shows.
Their numerical results seem to indicate the presence of quadratic effects on several parameters, which the authors fail to explain satisfactorily.
The statement of lines 276-7 is technically flawed. The rumen is the principal organ responsible for the absorption of VFA’s, and it has NO mucosa.
The statement on lines 307-8 is not what their data seem to show, i.e. NNH3 levels in rumen fluid were increased, not decreased.

Additional comments

The English of the paper needs to be revised thoroughly.
The reference list needs to be revised: some items are missing (line 99), others are incorrect (line 402); there are inconsistencies in the titles of the journals (use of capital letters, abbreviations).
There are several misconceptions that need to be corrected.

·

Basic reporting

The submission uses a clear, unambiguous and professional English.

The background/context provided is sufficient BUT literature references have some errors: references in lines 402-403 and 438-441 are not cited in the text.

The submission is self-contained with relevant results to hypotheses.

It has professional structure and raw data shared BUT there are some errors in FIGURES AND TABLES:
Lines 231-233 describe Table 5, when they should refer to Table 4, since Table 5 DOES NOT EXIST.
Figure 3 needs a better explanation.
Figure 5 is NOT DESCRIBED in the text, only in the discussion and these results are fundamental for the conclusions of the work.
On line 316, they talk about Table 5 when they should refer to Figure 5.
In Table 2, the word "control", in the corresponding column, is cut off.
It would be advisable to write the expressions in vivo, in vitro and et al, in italics.

Experimental design

No comment

Validity of the findings

No comment

---

## Round 0.2 · Minor Revisions

The paper has improved. Please submit a revised version considering the minor corrections from the reviewers.

Reviewer 1 ·

Basic reporting

No comments

Experimental design

No comments

Validity of the findings

No comments

Additional comments

The authors addressed well all comments from my review report. The revised manuscript is thus now ready to publish on PeerJ. Thank you for your substantial effort in the revision.

Reviewer 2 ·

Basic reporting

no comment

Experimental design

no comment

Validity of the findings

no comment

Additional comments

The manuscript was very much improved from the former version, and this reviewer considers it suitable for publication.

Just two last small changes the should be performed:

Line 74: NMR is not a simple “manipulation”, but rather a simple method or technique.

Line 420: this reference should be moved to line 484

·

Basic reporting

The submission uses a clear, unambiguous and professional English.

Errors that I had detected in the bibliography (lines 402-403 and 438-441 in original manuscript) were corrected, now references are correctly cited in the text.

The reference to Table 5 (non-existent) was deleted, to replace it with "Table 4".

Explanation provided in the letter for Figure 3, is more clear.

Other typographical errors were also corrected.

Experimental design

No comment

Validity of the findings

No comment

Additional comments

It seems to me that the comments of the reviewers were attended and that the presentation of the manuscript was considerably improved.

---

## Round 0.3 · accepted · Accept

Dear author

I can read that you have addressed all the reviewers concerns. The reviewers comments have been responded adequately.

I congratulate you for the nice piece of work, which will add value to PeerJ.